# Perceptions of tourists of the resources, ecological service functions and recreation value of the Guanwu National Forest Recreation Area

Shou-Tsung Wu[1], Yi-Ta Hsieh[2]*, Shang-Chuan Huang[3], Chun-Hung Wei[4], Chaur-Tzuhn Chen[4], Jan-Chang Chen [4]*

1 Department of Tourism Management, Shih Chien University, Kaohsiung, Taiwan, 2 General Research Service Center, National Pingtung University of Science and Technology, Neipu, Taiwan, 3 Graduate Institute of Bioresources, National Pingtung University of Science and Technology, Neipu, Taiwan, 4 Department of Forestry, National Pingtung University of Science and Technology, Neipu, Taiwan

* blue90234570@mail.npust.edu.tw (YTH); zzzjohn@mail.npust.edu.tw (JCC)

**Data Availability Statement:** All relevant data are within the manuscript and its Supporting Information files.

## Abstract

This study surveyed visitor perceptions of the resources of the Guanwu National Forest Recreation Area (GNFRA) in Taiwan', their perceptions toward the ecosystem service functions of the GNFRA, their expectations and opinions on its management, and their recreational activities. Independent sample t-tests and one-way analysis of variance (ANOVA) were used to examine the influences of the backgrounds of respondents on their perceptions toward the ecosystem service functions of the GNFRA. The importance-performance analysis (IPA) method was used to explore evaluations by tourists on the management of the GNRFA. Finally, a global positioning system (GPS) was used to process records of recreational activities by tourists within the GNFRA to understand the movement of tourists during their recreational activities within the GNRFA. The results showed that visitors to the GNRFA regarded its recreational resources to be valuable and that they had a high willingness to revisit. The visitors also showed a certain understanding of the ecosystem services provided by the forest ecosystem. There were significant differences in perceptions of ecosystem services among visitors of different backgrounds. In addition, the results of IPA showed the "importance" of perceptions and "performance" within the opinions of tourists on the management of the forest recreation area. The records of recreational activities by tourists showed that they experienced the most benefits when experiencing picturesque scenery along walking trails. The results of this study can contribute to future management of the GNRFA.

## Introduction

Taiwan has abundant forest resources, and a national forest recreation area (NFRA) has been established by the forestry bureau of the Council of Agriculture of the Executive Yuan. This

**Funding:** The authors received no specific funding for this work.

**Competing interests:** The authors have declared that no competing interests exist.

area is rich in animal and plant species and provides a good recreational environment, receiving many millions of tourists each year who participate in ecological tourism, outdoor leisure activities, and environmental education services [1]. There has been a rise in demand for tourism and recreation, thereby increasing the need for management of the NFRA to improve the quality of tourism, meet the diversified outdoor recreational needs of tourists, provide a high-quality ecological tourism experience, and strengthen environmental education. For this reason, the present study aimed to explore the evaluations of natural resources of the NFRA by tourists, their perceptions of ecosystem service functions, their recreation activities in the NFRA, and their perceptions of management of the NFRA.

Owing to their multiple functions, forest ecosystems play a key role in environmental protection. Previous studies have categorized forest ecosystems functions into ecological services and habitat, wealth, and social culture [2,3], or use and non-use values [4].Wu et al. [5] similarly emphasized that natural assets play an irreplaceable role in human well-being and that forest ecosystem services involve interrelationships between natural assets, energy, and information flow, which is a conceptual mechanism providing benefits and promoting well-being. Based on the above concepts, the present study first explored the evaluations of the resources of the NFRA by tourists and their perceptions of ecosystem service functions.

The services provided by the NFRA can be placed into several categories, and the quality of these services can decide the success or failure of operation and management [6]. Since the services provided by the NFRA will affect the quality of activities, some scholars have used recreation opportunity spectrum (ROS) to divide the recreation environment into three categories [7]: (1) natural environment attributes; (2) social attributes of tourist interactions, and; (3) environmental conditions provided by managers. This concept has integrated the supply side and demand side of recreational activities, allowing operators to plan products and improve their quality, thereby allowed tourists to enjoy the expected environment and recreational opportunities [8]. Therefore, the present study also exploring expectations and perceptions of the management of the NFRA among tourists.

Past studies on the perceived value of recreation experiences by tourists have mostly utilized questionnaire surveys [9–11], while a few studies have analyzed the behavior of tourists based on their movement. However, the widespread availability of global positioning system (GPS) technology has facilitated the recording of movement of tourists and their time spent at attractions within the NFRA as another focus of the present study. The results of this analysis can evaluate the feasibility of GPS as well as provide operation references for relevant recreational facilities.

In summary, the present study focused on the Guanwu National Forest Recreation Area (GNFRA) under the jurisdiction of the Hsinchu Forest District Office of the Forestry Bureau and the visitors to the GNFRA. The present study examined the above issues based on relevant theories combined with statistical analysis, and had the following objectives:

1. To understand the socio-economic background and characteristics of tourists to the GNFRAG, as well as their evaluations of the GNFRA resources and their perceptions of forest ecosystem service functions.

2. To analyze whether there are differences in the perceptions of forest ecosystem service functions among tourists with different backgrounds.

3. To explore the perceptions of tourists of the "importance" and "performance" of the management of the GNFRAG.

4. To identify the routes and time spent at attractions by tourists to the GNFRAG.

The results of the present study can act as a reference for development of ecotourism and the formulation of management strategies for forest recreation areas of Taiwan in the future.

## Literature review

### Travel motivation and destination choice

The factors influencing how people choose travel destinations are extremely complex and research into travel motivation has been an important field in tourism research [12]. Motivation for a travel destination choice is a dynamic concept that may vary from person to person [13]. Therefore, there have been many empirical approaches applied to the concept of motivation as an element of tourism market segmentation [14,15]. In fact, tourist decision-making processes are complex and multifaceted [16] and are affected by interactions between several psychological (internal) and non-psychological (external) variables manifesting at different stages [17]. Therefore, some scholars have suggested several important criteria for selecting related issues for exploration [18].

The "push and pull" model is often used to explore the role played by travel motivation in destination choice [19]. The model draws from seven socio-psychological ("push") motivations, namely escape, self-exploration, relaxation, prestige, regression, kinship-enhancement, and social interaction, and two cultural ("pull") motivations, namely novelty and education. The model has been refined and corrected by subsequent studies by summarizing, internal ("push") and external ("pull") motivators for travel. Internal motivators include desire for escape, rest, relaxation, prestige, health and fitness, adventure, and social interaction. External motivators are based on the attractiveness of the destination, including tangible resources, and the perceptions and expectations of tourists [20]. The present study focuses on a forest recreation area categorized as a nature tourism destination. The resources of the recreational area show strong external attraction and can be categorized as "pull" motivations, thereby making it possible to introduce this concept for resource evaluation.

### The ecosystem service functions and recreation attractions of the forest recreation area

"Ecosystem functions" refers to functions provided by the ecosystem, whereas "ecosystem services" are the tangible or intangible (direct or indirect) benefits provided by the ecosystem to humans [21]. Related studies have found that biodiversity improves ecosystem functions (e.g., primary production, decomposition, nutrient cycling, and nutrient function), thereby supporting a wide range of ecosystem services (e.g., food production, climate regulation, pest control, and assimilation of pollination) [22]. Forests cover a large area in Taiwan and forest ecosystems contain abundant natural assets and biological resources. Hence, forest ecosystems can provide considerable service functions. However, their dynamic characteristics are difficult to accurately characterize due to the high degree of complexity and long-life cycle of ecosystem service functions [23]. Meanwhile, forest ecosystems exhibit multiple functions, resulting in different human development approaches. Therefore, the evaluations of forest ecosystem service functions are of great significance.

The satisfaction of individual tourists when visiting forest recreation areas is closely related to their experience and how they perceive the environment. However, perceptions of the environment differ among visitors to forest recreational areas due to their different interests in the environment. Therefore, some studies have considered that the recreation experience is a particular specific psychological feeling resulting from the participation in recreational activities

[24] and regard the motivation for recreational activities as a key determinant of the behavior of tourists [25].

However, the factors affecting the attraction of recreation activities are quite complex, including tourist sentiment [26], brand equity [27], the perceived image of the destination [28], cultural proximity [28], familiarity with the destination [29], facility allocation [30], environmental conditions [31], and the adopted management strategy [24]. In other words, the factors affecting the attraction of recreation activities are complex and pluralistic and cannot be resolved through the investigation of any single factor. Therefore, the present study only explored the associated business management issue.

NFRAs provide a suitable outdoor recreation area for enhancing the opportunities for tourists to interact with each other in harmony, often resulting in profound experiences. The degree to which expectations of green space by visitors are realized can significantly affect their perceptions and satisfaction, and even positively influence behavioral intentions [32]. However, since the management of forest recreation areas will affect artificial development and natural conservation within NFRAs, the study of relevant aspects of their management is worthwhile. For example, one study emphasized the importance of five factors, namely space, nature, culture and history, quietness, and facilities, within the assessment of the quality of green space services in parks by visitors [33]. To sum up, the present study proposes that the key to maintaining service quality and promoting the recreation experience of tourists is to explore the perceptions and opinions of tourists of management of recreation areas.

## Research methods

### Survey area and research subject

The survey area of the current study encompasses the 26th, 27th, and 28th forest compartments of the Zhudong working circle and the 48th forest compartment of the Da-An river working circle under the jurisdiction of the Hsinchu Forest District Office of the Forestry Bureau, with a total area of 907.42 ha. The present study selected the forest recreation areas as the survey areas and tourists visiting these areas as the survey subjects.

### Questionnaire design and sampling survey

The study areas were first visited to conduct a recreation environment survey and an interview with a local expert, following which a structured questionnaire was designed based on the interview, survey results, and relevant literature [34,35]. The questionnaire was divided into four parts: (1) tourism behaviors and their evaluations of natural resources; (2) the perceptions of tourists of forest ecosystem service functions; (3) opinions of tourists of the operation and management of NFRAs; (4) the backgrounds of tourists (the questionnaire is in the supplemental information). Among these parts, the first part aimed to obtain an understanding of the behavior of tourists and their evaluation of resources, incorporating a total of 11 questions. The second part mainly aimed to obtain an understanding of the perceptions of tourists towards forest ecosystem service functions, with a total of 21 questions and utilizing the Likert five-level scale of "highly disagree" to "highly agree", providing a grading of 1 to 5, respectively. The third part mainly aimed to identify the satisfaction of tourists of the current operation and management of the NFRAs, incorporating a total of 30 questions, also using the Likert five-level scale of "highly unimportant" to "highly important", thereby providing a grading of 1 to 5, respectively. Finally, the fourth part aimed to obtain information on the background attributes of the surveyed subjects, including gender, age, educational background, marital status, occupation, monthly income, monthly working days, residence, and other attribute data, incorporating a total of 8 questions.

The present study selected tourists to the GNFRA as survey subjects. Considering the number of local visitors (https://stat.taiwan.net.tw/scenicSpot, 2021/06/10) and referring to the relevant literature [36,37], the questionnaire was distributed using the convenience sampling method after interviewer training and confirmation of relevant process, considering both weekdays and holidays. The main scope of the interviews incorporated the tourist center, footpaths, and scenic spots. The interviewers visited the survey area from January to April, 2017 during which they distributed questionnaires to willing respondents. A total of 500 completed questionnaires were collected (statistics by interviewers indicated a response rate of 55%) and 28 invalid questionnaires were deducted after sorting, resulting in a total number of valid questionnaires of 472. The Hsinchu Forest District Office authorized the questionnaire survey conducted by the present study. In addition, the management office of the Guanwu National Forest Recreation Area provided verbal consent prior to the questionnaire survey. All interviewees provided consent prior to taking part in the interview. When interviewees were minors, verbal consent was obtained from accompanying parents before the interview was conducted.

## Data processing and analysis

After the design of the questionnaire was finalized, the questionnaire was evaluated by other teachers in the Department of Forest and Tourism. Considering suggestions provided during the evaluation, the questionnaire was revised and the contents of the survey were deemed valid. No items were deleted after item analysis. The responses relating to the perceived importance of forest ecosystem service functions and the management of forest recreation areas and their satisfaction with these elements all passed the Kaiser-Meyer-Olkin (KMO) test, the measure of sampling adequacy, and the Bartlett's test (KMO = 0.935, 0.929, and 0.935, respectively; p = 0.000), indicating that good sampling appropriateness was achieved for the above items. The reliability of responses was confirmed by the Cronbach's α values of 0.958, 0.961, 0.955, respectively, indicating consistent responses between respondents. After testing the reliability and validity of the scale data, descriptive statistics were used to analyze the respondent background attributes, tourist behavior, the evaluations of recreation resources by tourists, and the respondent perceptions of forest ecosystem service functions. Statistical analyses in the present study were conducted in the IBM® Statistical Package for the Social Sciences (SPSS®) software.

The questionnaire items were arranged into four major functions to identify differences among the perceptions of forest ecosystem service functions by tourists with different backgrounds: (1) supply; (2) regulation; (3) support, and; (4) culture. Each functional dimension was represented by their average value. The independent sample t-test and one-way analysis of variance (ANOVA) were then used to explore whether tourists of different backgrounds showed differences in their perceptions of forest ecosystem service functions. Finally, the importance-performance analysis (IPA) [38] method was adopted to explore the evaluation of current forest recreation area management by tourists.

## Analysis of the recreational activities of visitors to the GNFRA

The present study distributed hand-held GPS systems to consenting tourists so as to characterize the movement of tourists during recreation. The GPS system used was the Garmin eTrex® 30x hand-held satellite positioning instrument. During the tracking of tourist movement, one coordinate point was automatically recorded every 30 seconds. The time of day of each position was also recorded to analyze the time spent by tourists at individual attractions in the park.

## Results

### Tourist background

The present study analyzed tourist backgrounds using the frequency allocation table (Table 1). As shown in Table 1, male and female tourists account for 47.2% and 52.8% of total tourists, respectively, with female tourists slightly outnumbering male tourists. The age structure of tourists showed that 59.2% of the tourists were over 40 years of age, indicating that the recreational characteristics of this area might be more attractive to middle-aged and elderly people. The educational background of survey participants showed that 54.2% of tourists have a college degree or above, indicating that the general public have a high level of education. A total of 62.5% of respondents were married, which is likely related to the age distribution of respondents. The northern ethnic group accounted for the majority (60.2%) of survey respondents, indicating that most visitors to the recreation area originate from neighboring areas. Therefore, scope remains for strengthening the publicity of the NFRAs. There was a wide distribution of occupation, monthly income, and monthly working days among respondents, indicating that the present study obtained a representative sample of tourists.

### Tourism behaviors and their evaluations of the resources of recreational areas

The present study analyzed tourism behaviors and their evaluations of the resources of the recreation area using the frequency distribution table (Fig 1). As shown in Fig 1, among the

**Table 1. Background attribute frequency allocation table of tourists visiting the Guanwu National Forest Recreation Area in Taiwan.**

| Background attributes | Sub-item | Percentage (%) | Background attribute | Sub-items | Percentage (%) |
|---|---|---|---|---|---|
| Gender | Male | 47.2 | Occupation | Students | 15.7 |
| | Female | 52.8 | | Soldiers, civil servants, teachers | 7.8 |
| Age | Under 20 | 6.8 | | Agriculture, forestry, fishery, animal husbandry and mining | 1.1 |
| | 21–30 | 20.6 | | Commerce | 12.1 |
| | 31–40 | 13.6 | | Industry | 13.6 |
| | 41–50 | 28.0 | | Service industry | 16.7 |
| | 51–60 | 22.5 | | Flexible job | 12.5 |
| | Above 61 | 8.7 | | Retirees | 12.1 |
| Educational background | Elementary school | 0.8 | | Others | 8.5 |
| | Junior high school | 7.4 | Monthly income (NTNT$) | Below 20,000 | 25.2 |
| | Senior high school (Vocational high school) | 19.1 | | 20,001–30,000 | 16.1 |
| | Junior college | 18.4 | | 30,001–40,000 | 15.7 |
| | College | 39.8 | | 40,001–50,000 | 12.5 |
| | Above graduate school | 14.4 | | 50,001–60,000 | 7.6 |
| Marital status | Unmarried | 36.4 | | 60,001–70,000 | 8.9 |
| | Married, no children | 23.7 | | 70,001–80,000 | 6.4 |
| | Married, with children | 38.8 | | 80,001–90,000 | 2.5 |
| | Others | 1.1 | | 90,001–100,000 | 0.8 |
| Residence | North Taiwan | 60.2 | | Above 100,000 | 4.2 |
| | Middle Taiwan | 12.7 | Monthly working days | Below 10 days | 20.1 |
| | South Taiwan | 26.7 | | 10–15 days | 7.8 |
| | East Taiwan | 0.4 | | 16–20 days | 25.8 |
| | Offshore island | 0.0 | | 21–25 days | 40.0 |
| | Foreign | 0.0 | | Above 26 days | 6.1 |

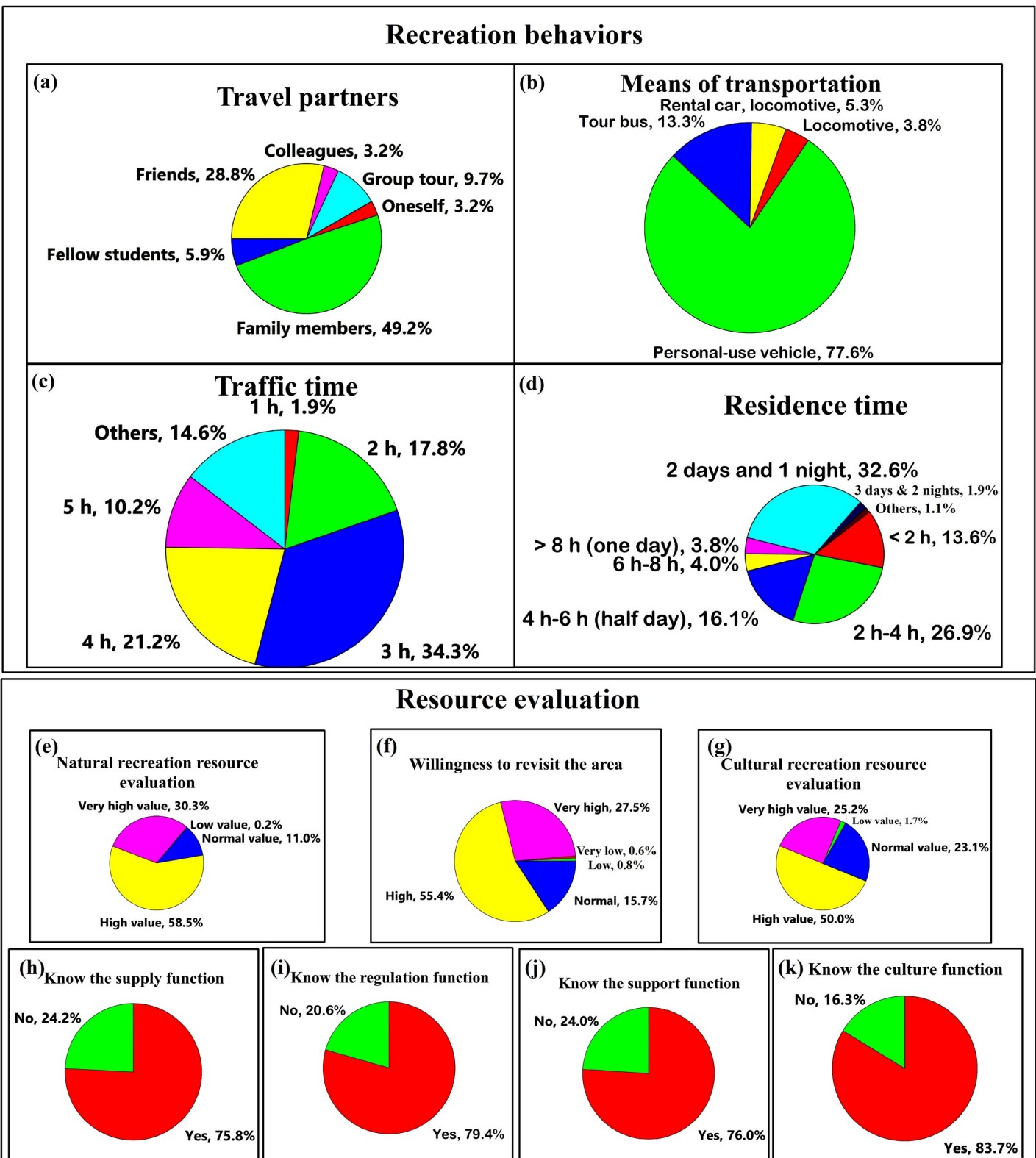

**Fig 1. The behavior and evaluations of resources by tourists visiting the Guanwu National Forest Recreation Area in Taiwan.** Data for this analysis were divided into two parts: "recreation behaviors" and "resource evaluation"; recreation behaviors included information on travel partners, means of transportation, traffic time, and residence time; the resource evaluation included data for tourist evaluations of the natural and cultural recreation resources, willingness to revisit the area, and awareness of the supply, regulation, support, and culture functions.

companions of respondents, the highest percentage was family members (49.2%), followed by friends (28.8%), thereby indicating that the NFRAs provide a good environment for relatives and friends to visit together, and also characterizes the tourism habits of visitors to the NFRAs. Private cars accounted for the vast majority (77.5%) of means of transport to the GNFRA among respondents, which shows diversification of transportation modes among visitors to the NFRAs and highlights potential parking space shortages that should be considered within GNFRA management. The travel time to the GNFRA of 65.7% of the tourists was 3 h to 5 h, indicating that the distance travelled by most respondents to the survey was considerable; therefore, the maintenance of road networks leading to the GNFRA is worthy of attention. The percentages of tourists spending 2 days, 1 night and 2 h to 4 h were 32.6% and 26.9%, respectively, indicating that tourists to the NFRAs may make a day trip to the GNFRA or they may stay overnight.

The evaluation of GNFRA resources by respondents showed that tourists attributed high or extremely high value to the natural and cultural recreational resources (88.8% and 75.2%, respectively) of the GNFRA; therefore, the NFRAs can be regarded as an attractive option for outdoor recreation. There is potential for the NFRAs to develop environmental and mountaineering education programs since tourists indicated high or very high willingness to revisit the GNFRA (82.8%). Over 70% of tourists recognized the supply, regulation, support, and cultural functions of forest recreation areas; therefore, management of NFRAs should strive to maintain normal operations of the national forest recreation areas so as to maintain their positive function and benefits to the public.

## Perception of forest ecosystem service functions among tourists

Furthermore, the present study analyzed the perceptions of forest ecosystem service functions in the recreational areas among respondents, and the average and standard deviation of each function item were calculated (Fig 2). As shown in Fig 2, the respondents generally expressed that the value of ecosystem service functions of the forest recreation areas were quite high, with the average scores of most categories of ecosystem functions above 4 (except the provision of timber, food and fresh water, and medical resources). In particular, the ecosystem service categories of providing a relaxing environment (4.39), the regulation of local climate and air quality (4.38), preventing soil erosion and maintaining soil fertility (4.37), maintaining species diversity (4.37), providing the necessary environment for the survival of animals and plants (4.35), establishing buffer zones (such as the slope stabilizing function of trees) to prevent natural disasters (4.35), and maintaining gene diversity (4.34) obtained high average scores.

The independent sample t-test and one-way ANOVA were then used to analyze whether tourists of different backgrounds showed differences in their perceptions of forest ecosystem service functions, with the results shown in Table 2. As shown in Table 2, there were significant differences in the responses according to respondent gender and occupation. There were significant differences in responses on the importance of regulation and supporting functions of ecosystem services according to the age of respondents. The importance placed on the regulating, supporting, and cultural functions of ecosystem services differed among respondents according to educational background. The importance placed on the supply function of ecosystem services differed among respondents according to marital status and residential area. The differences between respondents in monthly income appeared to have a significant effect on the value they attributed to the regulating and cultural functions of ecosystem services.

## Evaluations of the management of forest recreation areas among tourists

IPA was first proposed by Martilla and James [38] and draws the average score of importance and performance into a two-dimensional matrix to allow an analysis of their correlation. The

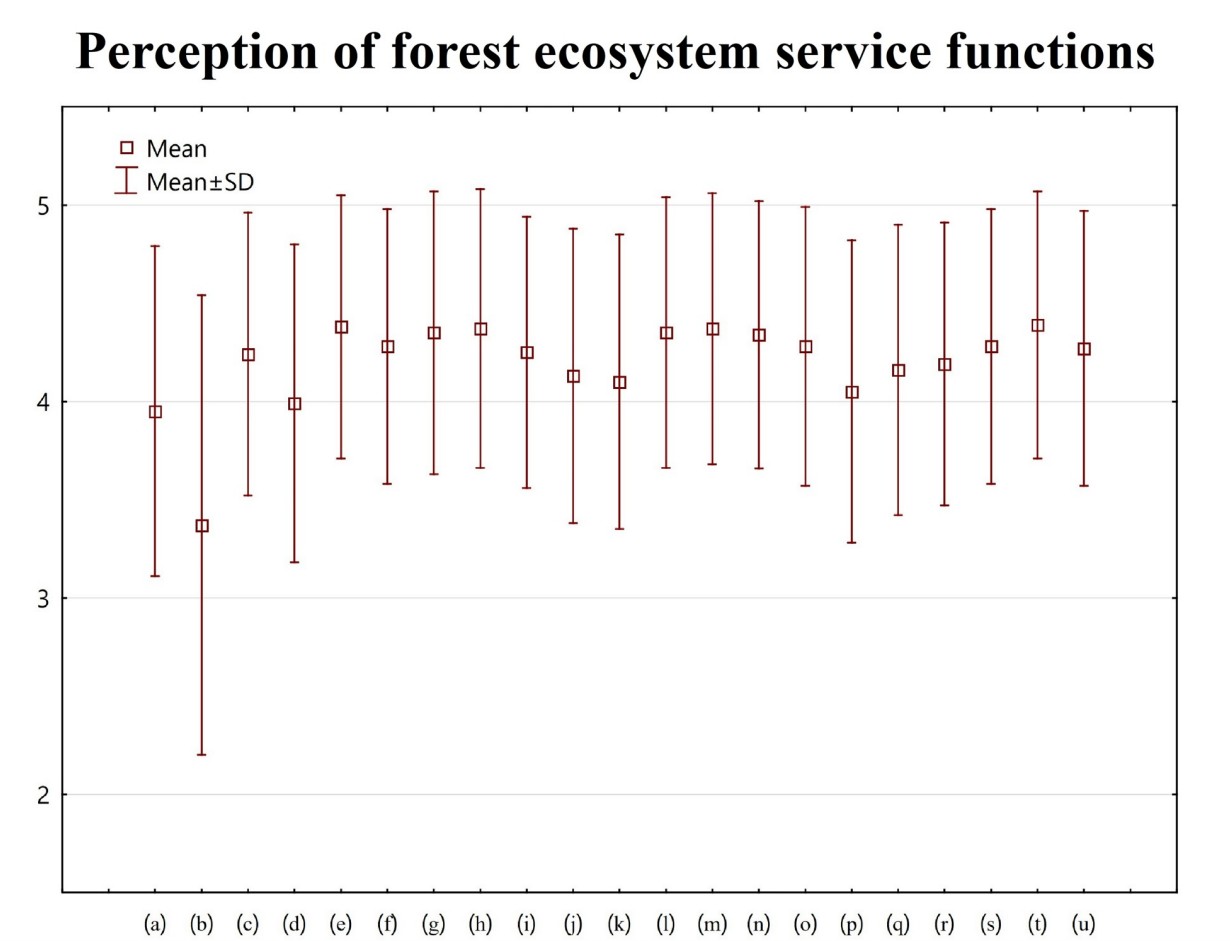

(a)Able to provide food, fresh water, etc. ; (b)Able to provide forest production; (c)Play the role in regulating flow and purifying water quality; (d)Able to provide medical resources;(e)Able to regulate the local climate and air quality; (f)Carbon storage and greenhouse gas reduction; (g)Establishes a buffer zone (such as the slope stabilizing function of trees) to prevent natural disasters; (h)Prevent soil erosion and maintain soil fertility; (i)Able to filter and decompose waste; (j)Provide pollination; (k)Pest control; (l)Provide necessary environment for animals and plants to survive; (m)Maintain species diversity; (n)Maintain genetic diversity; (o)Maintain the diversity of ecosystems; (p)Influence on art and culture; (q) Provide a leisure function; (r) Contribute to the academic field; (s)Provide spaces for environmental education; (t)Provide a relaxing environment; (u) Maintaining a sense of belonging between people and the natural environment; SD: standard deviation.

**Fig 2. Results of the analysis of the perceptions of forest ecosystem service functions by tourists to the Guanwu National Forest Recreation Area in Taiwan.**

four quadrants of an IPA matrix are continuous maintenance, oversupply, low priority, and enhanced improvement [39].

The current study used IPA to extract the spatial distribution of importance-performance to understand the importance of perceptions of tourists of the management of the GNFRA, with the results shown in Fig 3. As shown in Fig 3, the tourists were satisfied with the number of public facilities (parking lots, public toilets, service centers, etc.), recreation route planning, the attitudes of park staff, noise control, waste (garbage) treatment, water quality and water source protection, air quality, suitability of building facilities for merging with the natural landscape, sustainable development of local culture, conservation of the natural ecology, assessment of recreation capacity, monitoring of the overall environment, relaxation value, and the cognition of natural ecology. It is recommended that management of GNFRAG should

**Table 2. The results of analysis of variations in the value placed on ecosystem service functions among tourists to the Guanwu National Forest Recreation Area in Taiwan according to tourist background.**

| Background attribute / Function category | gender | Age | Educational background | Marital status | Occupation | Monthly income | Residential area |
|---|---|---|---|---|---|---|---|
| | t value df | F value df | | | | | |
| Supplying function | 2.522* 468.46 | 2.101 471 | 2.228 471 | 3.071* 471 | 2.219* 471 | 1.457 471 | 4.178** 471 |
| Regulating function | 2.464* 457.16 | 3.696** 471 | 5.283** 471 | 0.739 471 | 3.906** 471 | 3.015** 471 | 1.083 471 |
| Supporting function | 3.113** 437.76 | 3.781** 471 | 4.878** 471 | 0.061 471 | 2.542* 471 | 0.888 471 | 1.179 471 |
| Culture function | 3.091** 455.84 | 1.031 471 | 4.211** 471 | 1.062 471 | 5.000** 471 | 2.814** 471 | 0.268 471 |

* At the significant level of 0.05 (two tailed), the difference is significant

** at the significant level of 0.01 (two tailed), the difference is significant.

continue to maintain these aspects. Tourists were dissatisfied with park signage (fold-out brochures, commentary boards, sign boards, etc.) and advocacy of environmental education. It is recommended that management of GNFRAG strengthen these aspects. Tourists showed little concern or general satisfaction with traffic convenience (public transportation, roads, etc.), accessibility of food and accommodation, number of service personnel, external publicity and marketing, participation of local residents, control of the number of tourists, planning of activities (field experience, etc.), arrangement of interpretation services, recreation connections with nearby scenic spots, the selling of commodities (books, souvenirs, etc.), and improvement of interpersonal relationships. Therefore, there is a low priority for the improvement of these aspects in the GNFRAG. Tourists were also satisfied but not concerned with the promotion of

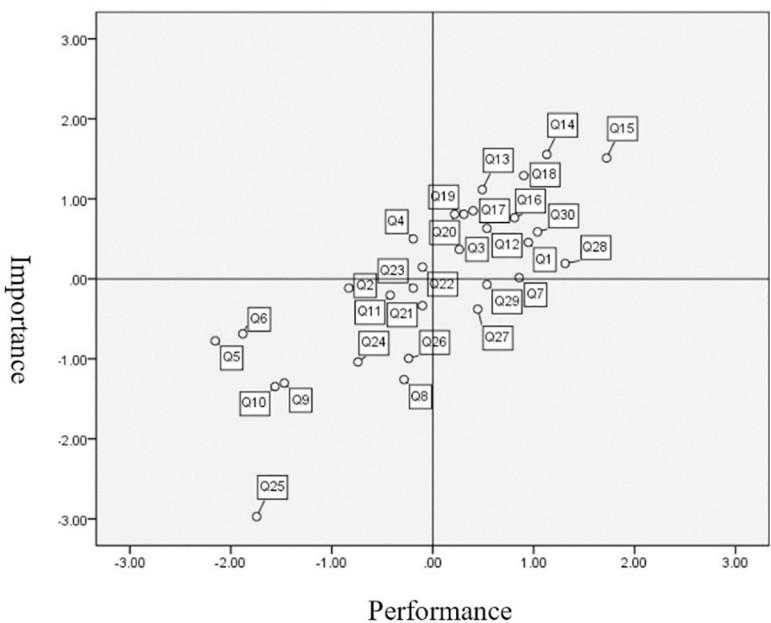

**Fig 3. Results of importance-performance analysis for management aspects within the Guanwu National Forest Recreation Area in Taiwan.** Q1 to Q30 can be matched with the questions in "Part III Operating Management Topics" in the questionnaire (see supporting information).

**Table 3. The analysis results of the visiting times and preferred pathways and facilities within the Guanwu National Forest Recreation Area in Taiwan by tourists.**

| Main analysis items | Item description | Percentage % | Main analysis items | Item description | Percentage % |
|---|---|---|---|---|---|
| Start time of recreation | Before 10 a.m. | 62 | Footpath preference | Waterfall footpath | 23 |
| | After 10 a.m. (including afternoon) | 38 | | Huishan footpath | 41 |
| Footpath preference | Waterfall footpath | 30 | | Zhenshan footpath | 11 |
| | Facilities in upper half of Huishan footpath | 38 | | BirdBird-watching footpath | 2 |
| | BirdBird-watching footpath | 2 | | Yunwu footpath | 23 |
| | Yunwu footpath | 30 | | | |

family harmony and cultural preservation. Therefore, it can be concluded that these aspects are well represented within the GNFRAG.

## Recreational activities of tourists

The present study obtained the consent of 42 tourists for wearing the GPS systems. The data collected were processed and plotted in Arc GIS, with the results of analysis shown in Table 3, Figs 4 and 5. As shown in Table 3, most tourists entered the GNFRA before 10 a.m., with fewer arriving in the afternoon, with most of the latter group staying overnight, which is likely related to the park opening time (8 a.m. to 5 p.m.). Data for the footpath preferences of tourists showed that five main footpaths in the GNFRA were used. Most tourists preferred to use the Huishan footpath, possibly due to the presence of five iconic giant red cypress trees of around a thousand years old and the ease of access the gentle slope of the footpath provides, thus serving as the mass tourism route. The data for the length of time spent along different paths showed that a higher proportion of tourists visited attractions along the Huishan, Waterfall, and Yunwu footpaths, indicating that tourists utilized these facilities at a high rate.

As shown in Fig 4, the range of distance available for recreational activities offered to tourists in the GNFRA is 0.21 km to 28.41 km, with an average of 11.21 km. However, the analysis of the GPS data showed that the H32 tourists deviated from the footpath when completing the Huishan footpath to attempt to visit the military radar station, which lasted about 5 h. These particular tourists (young women aged between 20 and 30) mentioned during returning the

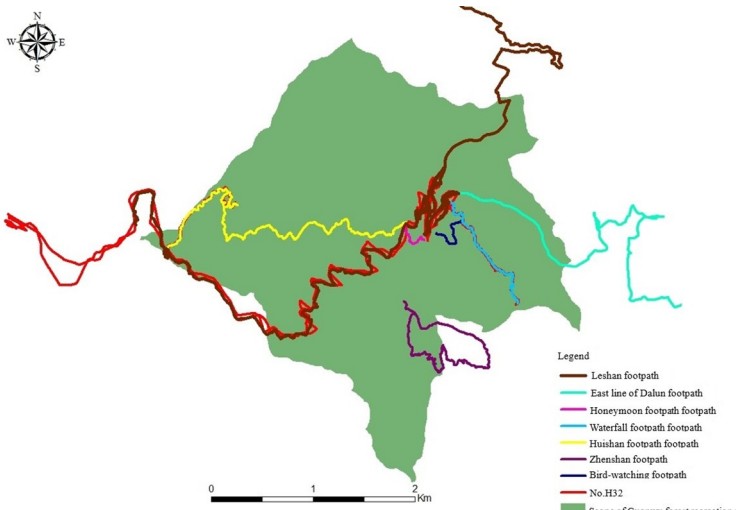

**Fig 4. Map showing the recreational activities of tourists to the Guanwu National Forest Recreation Area in Taiwan.**

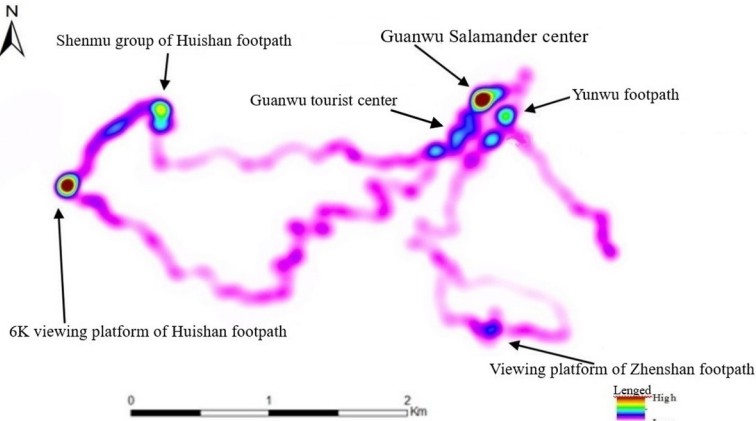

**Fig 5. Map showing the distribution of tourists visiting attractions within the Guanwu National Forest Recreation Area in Taiwan.**

GPS that: "A route exists to drive after the 6K viewing platform on the Leshan forest road, and since there was no sign prohibiting access, we proceeded to enter. However, we were later unexpectedly turned back by a military checkpoint". Although this represents a scarce incident, it suggests that the management of the GNFRA should set up warning signs at necessary places for the safety of tourists.

The popularity of park attractions was assessed according to the time spent at these areas by tourists (Fig 5). Within Fig 5, shades of the area indicate the degree to which they are visited by tourists. Tourists parked at the Guanwu tourist center and requested information at the Guanwu Salamander center, and therefore these areas showed higher rates of usage. Other attractions receiving high traffic were the Shenmu group, the 6K viewing platform of the Huishan footpath, the Yunwu footpath, and the viewing platform of the Zhenshan footpath.

## Discussion

The results of the analysis conducted in the present study indicate the need for further discussion of the points shown below.

The analysis of tourist behavior and evaluation of resources indicates that recreational resources can be divided into different types, including natural environment, service facilities, and recreational equipment [40]. The forest recreation area (GNFRA) examined in the present study can be described as a mountainous and wilderness area with relatively few complete accommodation, dining, and entertainment facilities. From the perspective of tourist behavior, for most tourists, travel to the GNFRA requires a long journey, and most tourists make this journey by private car. Therefore, many tourists will want to stay overnight. This highlights the importance of upgrading accommodation, dining, and entertainment facilities, and for maintaining a good quality road network and providing sufficient parking space. In addition, NFRAs are characterized by many unquantifiable public benefits in addition to the economic benefits of providing recreation [41]. The rich natural resources of the study area are indeed recognized by tourists, consistent with the results of other similar studies [35]. The background of visitors may influence their evaluation of different types of ecosystem services.

Past studies mostly placed forest ecosystem service functions into three categories, namely ecological service and habitat functions, property provisioning functions, and socio-cultural functions [42]. The results of the present study highlighted the positive attitudes of tourists toward the four major functions of the GNFRA of supply, regulation, support, and culture.

Therefore, forest ecosystems in Taiwan do fulfil certain service functions. The present study identified differences in perceptions of service functions of forest ecosystems among different visitor backgrounds consistent with those of previous studies [43]. Although the analysis of tourist opinions on the importance of ecosystem service functions showed no consistent relationship with their background, other studies obtained similar results [35,43].

The challenges facing the management of forest recreation areas are diverse and complex. The present study, with reference to other related literature [44,45], used IPA to identify the "importance" of the perceptions of tourists and the "performance" of their opinions on the management of the GNFRA, which can be of substantial assistance to the subsequent development of the GNFRA.

Broach et al. [46] used GPS to recorded the activity of cyclists and evaluated their attraction to different types of facilities so as characterize which paths they select and why. Hallo et al. [47] similarly used GPS to study the spatial distribution and rate of activity of tourists to understand their use and preferences for different facilities among different regions. Meijles et al. [48] used GPS to analyze the recreation activities of tourists to a national park and estimated the rate of activity, distribution of activity, and preferred routes, with the results of their study helpful for characterizing the activity patterns of different hiking groups. Based on the above literature and results of the present study, the use of GPS is a convenient and accurate method for tracking the activity of tourists within recreation areas, and the data provided can relevant according to research topics or management needs.

A limitation of the present study was that it examined only one NFRA over a limited period of four months (January to April of a single year). Therefore, the results of the analysis remain practically limiting. Future studies could extend the current research to other NFRAs in Taiwan and incorporate further aspects. In addition, future studies could be conducted over longer durations. In this way, future studies can make a more substantial contribution to the development of ecotourism and environmental education in the NFRAs.

## Conclusions

The present study surveyed tourists to the GNFRA through a questionnaire survey and GPS records of tourist activities. Several important conclusions were made. First, from the perspective of tourism behavior and evaluation of resources, travelling to the GNFRA must take a long time, and most tourists travel to the GNFRA by private car. Therefore, there is value in upgrading accommodation, dining, and entertainment facilities, providing adequate parking space, and maintaining good road quality. Secondly, the respondents generally expressed that the recreation area provides good recreational resource value, and they expressed a high willingness to return. They also had certain perceptions of forest ecosystem service functions. Therefore, the management of the GNFRA should strive to maintain the normal operation of the park so as to maintain its existing value. Third, visitors from different backgrounds have different perceptions of forest ecosystem service functions. Therefore, the results of the present study suggests that it is reasonable to assume that the backgrounds of tourists will affect their valuations of different categories of ecosystem services. Fourth, the results of IPA showed that the "importance" of the perceptions of tourists of the management of recreation areas and the "performance" of their opinions can be characterized, which is helpful for improving the management of the recreation area. Fifth, the tracking of the recreational activities of tourists through the use of GPS not only clearly characterized their recreational activities and time spent at different attractions, but also effectively allowed an identification of the most popular scenic spots and common facilities in the region. Therefore, management of the GNFRA can make full use of these data according to the research topics or management needs.

NFRAs play an important role in the development of ecotourism and practical environmental education in Taiwan. There is a need to analyze the opinions of tourist on the operation and management of NFRAs and to study their recreational activities in these areas to improve the quality of services offered in the NFRAs. In this way, the NFRAs can meet the needs of tourist in terms of their perceptions on preferences for outdoor recreation, evaluations of NFRAs resources, and perceptions of ecosystem service functions. The results of the present study are helpful for understanding the behavior of tourists, their evaluations of NFRAs resources, and differences among tourists in their perceptions of ecosystem service functions according to their backgrounds. Briefly, the results of the present study are helpful for future development of the NFRAs in Taiwan.

## Supporting information

**S1 File. Questionnaire (Chinese).**
(DOCX)

**S2 File. Questionnaire (English).**
(DOCX)

**S1 Data. Guanwu data.**
(XLS)

**S2 Data. Guanwu data english.**
(XLS)

## Author Contributions

**Conceptualization:** Shou-Tsung Wu, Chaur-Tzuhn Chen, Jan-Chang Chen.

**Data curation:** Shou-Tsung Wu, Shang-Chuan Huang, Jan-Chang Chen.

**Investigation:** Shou-Tsung Wu, Yi-Ta Hsieh, Shang-Chuan Huang, Chun-Hung Wei.

**Methodology:** Shou-Tsung Wu, Yi-Ta Hsieh, Jan-Chang Chen.

**Supervision:** Chun-Hung Wei, Chaur-Tzuhn Chen, Jan-Chang Chen.

**Writing – original draft:** Shou-Tsung Wu.

**Writing – review & editing:** Yi-Ta Hsieh, Shang-Chuan Huang, Chun-Hung Wei, Jan-Chang Chen.

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
