## [Decision Letter · Decision Letter 0]

24 May 2021

PONE-D-21-12141

The study of evaluations of the resources of the Guanwu National Forest Recreation Area by tourists, their perceptions toward the ecosystem service functions provided and their recreational activities

PLOS ONE

Dear Dr.Chen,

Thank you for submitting your manuscript to PLOS ONE. After careful consideration, we feel that it has merit but does not fully meet PLOS ONE’s publication criteria as it currently stands. Therefore, we invite you to submit a revised version of the manuscript that addresses the points raised during the review process.

We look forward to receiving your revised manuscript.

Kind regards,

Shah Md Atiqul Haq

Academic Editor

PLOS ONE

Additional Editor Comments (if provided):

Dear authors,

I would like to ask you to revise the paper taking into account the comments and suggestions of the reviewers.

After I receive the revised version, I may send the paper again to new reviewers to get their opinion.

Yours sincerely,

Journal Requirements:

2)  Please include additional information regarding the survey or questionnaire used in the study and ensure that you have provided sufficient details that others could replicate the analyses. For instance, if you developed a questionnaire as part of this study and it is not under a copyright more restrictive than CC-BY, please include a copy, in both the original language and English, as Supporting Information.

3)  Thank you for including your ethics statement:  "A questionnaire survey was used for this study. Verbal consent was obtained from the

Guanwu National Forest Recreation Area Management Office prior to the questionnaire survey. Consent was sought from the subjects when the questionnaire was administered. Verbal consent was obtained from accompanying parents for all minor subjects.".   

4) We suggest you thoroughly copyedit your manuscript for language usage, spelling, and grammar. If you do not know anyone who can help you do this, you may wish to consider employing a professional scientific editing service.  

5) PLOS requires an ORCID iD for the corresponding author in Editorial Manager on papers submitted after December 6th, 2016. Please ensure that you have an ORCID iD and that it is validated in Editorial Manager. To do this, go to ‘Update my Information’ (in the upper left-hand corner of the main menu), and click on the Fetch/Validate link next to the ORCID field. This will take you to the ORCID site and allow you to create a new iD or authenticate a pre-existing iD in Editorial Manager. Please see the following video for instructions on linking an ORCID iD to your Editorial Manager account: https://www.youtube.com/watch?v=_xcclfuvtxQ

6) Please amend either the abstract on the online submission form (via Edit Submission) or the abstract in the manuscript so that they are identical.

7) We note that Figure 1 in your submission contain map/satellite images which may be copyrighted. All PLOS content is published under the Creative Commons Attribution License (CC BY 4.0), which means that the manuscript, images, and Supporting Information files will be freely available online, and any third party is permitted to access, download, copy, distribute, and use these materials in any way, even commercially, with proper attribution. For these reasons, we cannot publish previously copyrighted maps or satellite images created using proprietary data, such as Google software (Google Maps, Street View, and Earth). For more information, see our copyright guidelines: http://journals.plos.org/plosone/s/licenses-and-copyright.

i.    You may seek permission from the original copyright holder of Figure(s) [#] to publish the content specifically under the CC BY 4.0 license. 

ii.    If you are unable to obtain permission from the original copyright holder to publish these figures under the CC BY 4.0 license or if the copyright holder’s requirements are incompatible with the CC BY 4.0 license, please either i) remove the figure or ii) supply a replacement figure that complies with the CC BY 4.0 license. Please check copyright information on all replacement figures and update the figure caption with source information. If applicable, please specify in the figure caption text when a figure is similar but not identical to the original image and is therefore for illustrative purposes only.

Reviewers' comments:

Reviewer's Responses to Questions

**Comments to the Author**

1. Is the manuscript technically sound, and do the data support the conclusions?

Reviewer #1: No

Reviewer #2: Yes

2. Has the statistical analysis been performed appropriately and rigorously? 

Reviewer #1: No

Reviewer #2: Yes

3. Have the authors made all data underlying the findings in their manuscript fully available?

Reviewer #1: Yes

Reviewer #2: Yes

4. Is the manuscript presented in an intelligible fashion and written in standard English?

Reviewer #1: Yes

Reviewer #2: Yes

5. Review Comments to the Author

Reviewer #1: This study sought to examine the perceptions of tourists towards their use of a national forest recreation area (NFRA) in Taiwan. The authors implemented a questionnaire of 500 tourists and analyzed their responses using t-tests, ANOVA and importance-performance analysis (IPA). There was a nice summary of the literature, but the way it was presented read more like a list of information than a narrative. Though the research is interesting, and I see potential in this paper, I have several concerns about it that lead me to not accept it for publication. I do want to commend the authors for undertaking this work without any specific funding support, as I understand how challenging that can be.

Particular areas that need improvement include: further explanation of the methods, more technical reporting of results, and higher level interpretation of the implications of the results in the discussion. It also might be my personal preference, but I think the results should be separate from the discussion as it was difficult to distinguish between presentation and interpretation of results. I have gone into further detail about my specific recommendations below.

Methods:

I recommend including the entire survey questionnaire as supplemental information so as to assist with the replicability of the study and so readers can understand the variety of questions asked. Another recommendation I have is to provide more information about how the survey respondents were selected using convenience sampling, and include a citation to a reputable source that you used as a guide for the sampling (line 182). Please also include more information about what software(s) you used to generate the figures and conduct the statistical analyses. This will also aid in replicability. I am also curious about the response rates of the survey implementation, and what the authors did to reduce any bias in their sampling regime. About how many tourists visit the park each day and what proportion of that number was sampled? Did this study undergo any ethics review and how were the respondents given informed consent? There’s a lot missing here. I recommend reading more literature about quantitative survey design and presentation in natural resource management for guidance.

Results:

The paragraph beginning on line 251 seems like it would be better suited in the Methods as it describes evaluation of the survey questionnaire and analysis of the data. Lines 294-298 also should go in the Methods.

I recommend improved presentation of survey results so it’s easier to visualize trends. Table 1 does a nice job of demonstrating the demographic snapshot of respondents, and these types of data are usually depicted in a table. However, I think the data presented in Tables 2 and 3 would be better viewed in a figure rather than a table. R Studio has a great Likert package to assist with visualizing Likert-style data, but I think you can also make stacked bar plots in Excel in a similar fashion, overlaying the percent of respondents who gave each answer. The data presented in Table 4 should also include the degrees of freedom (df) used in the analyses. Another suggestion for analyzing responses could be a Chi-square analysis. Also, where are the data stored? Please include that information in the manuscript. Some public (and free, I think) places are GitHub and Data Dryad.

Discussion:

Overall, the discussion of the implications of the results and applicability to the literature read more like a list of previous works rather than fitting the main conclusions of this study within the literature. One major suggestion I have is to break up the discussion into its own section and to make sure each paragraph has its own point to make. Each point from the paragraph should apply to a key finding of the study and how it fits in with the literature. I also recommend ending the paper with a very broad conclusion, not the limitations of the study.

There is interesting information in here, but I think the paper falls short of presenting it in the best way to be understood. Depending on the responses of the other reviewers, I would be willing to read further versions of this manuscript.

Reviewer #2: Firstly, all tables need to changes to the histograms because the reader needs to look at the comparison between the groups clearly. the title needs more concise. The abstract rewrites and mentions all the results that the researcher got them. finally, The references need to update.

6. PLOS authors have the option to publish the peer review history of their article (what does this mean?). If published, this will include your full peer review and any attached files.

Reviewer #1: No

Reviewer #2: No

---

## [Author Response · Author response to Decision Letter 0]

19 Jul 2021

Response to Reviewers

Reviewers' comments:

Reviewer's Responses to Questions

Comments to the Author

1. Is the manuscript technically sound, and do the data support the conclusions?

Reviewer #1: No

Reviewer #2: Yes

2. Has the statistical analysis been performed appropriately and rigorously?

Reviewer #1: No

Reviewer #2: Yes

3. Have the authors made all data underlying the findings in their manuscript fully available?

Reviewer #1: Yes

Reviewer #2: Yes

4. Is the manuscript presented in an intelligible fashion and written in standard English?

Reviewer #1: Yes 

Reviewer #2: Yes

 

5. Review Comments to the Author

Reviewer #1: This study sought to examine the perceptions of tourists towards their use of a national forest recreation area (NFRA) in Taiwan. The authors implemented a questionnaire of 500 tourists and analyzed their responses using t-tests, ANOVA and importance-performance analysis (IPA). There was a nice summary of the literature, but the way it was presented read more like a list of information than a narrative. Though the research is interesting, and I see potential in this paper, I have several concerns about it that lead me to not accept it for publication. I do want to commend the authors for undertaking this work without any specific funding support, as I understand how challenging that can be.

Particular areas that need improvement include: further explanation of the methods, more technical reporting of results, and higher level interpretation of the implications of the results in the discussion. It also might be my personal preference, but I think the results should be separate from the discussion as it was difficult to distinguish between presentation and interpretation of results. I have gone into further detail about my specific recommendations below.

Respones:

We thank the reviewers for their valuable comments on this paper. The literature review and analysis methods have been supplemented according to the comments of the members. In addition, the results and the discussion have been written separately in this paper, and the presentation and explanation have been strengthened.

Methods:

I recommend including the entire survey questionnaire as supplemental information so as to assist with the replicability of the study and so readers can understand the variety of questions asked. Another recommendation I have is to provide more information about how the survey respondents were selected using convenience sampling, and include a citation to a reputable source that you used as a guide for the sampling (line 182). Please also include more information about what software(s) you used to generate the figures and conduct the statistical analyses. This will also aid in replicability. I am also curious about the response rates of the survey implementation, and what the authors did to reduce any bias in their sampling regime. About how many tourists visit the park each day and what proportion of that number was sampled? Did this study undergo any ethics review and how were the respondents given informed consent? There’s a lot missing here. I recommend reading more literature about quantitative survey design and presentation in natural resource management for guidance.

Respones:

1. Due to the length of the paper, the content of the survey questionnaire is supplemented in the "Research Results" section.

2. As suggested by the reviewers, we supplemented the literature on the lobbying data and sampling methods.

3. As suggested by the reviewers, the description of data analysis methods is enhanced.

Results:

The paragraph beginning on line 251 seems like it would be better suited in the Methods as it describes evaluation of the survey questionnaire and analysis of the data. Lines 294-298 also should go in the Methods.

I recommend improved presentation of survey results so it’s easier to visualize trends. Table 1 does a nice job of demonstrating the demographic snapshot of respondents, and these types of data are usually depicted in a table. However, I think the data presented in Tables 2 and 3 would be better viewed in a figure rather than a table. R Studio has a great Likert package to assist with visualizing Likert-style data, but I think you can also make stacked bar plots in Excel in a similar fashion, overlaying the percent of respondents who gave each answer. The data presented in Table 4 should also include the degrees of freedom (df) used in the analyses. Another suggestion for analyzing responses could be a Chi-square analysis. Also, where are the data stored? Please include that information in the manuscript. Some public (and free, I think) places are GitHub and Data Dryad.

Respones:

1. Some of the results were adjusted to the Research methods section according to the comments of the members. 2.

2. Some of the table contents are presented in graphical form.

3. Results and discussion are written separately.

Discussion:

Overall, the discussion of the implications of the results and applicability to the literature read more like a list of previous works rather than fitting the main conclusions of this study within the literature. One major suggestion I have is to break up the discussion into its own section and to make sure each paragraph has its own point to make. Each point from the paragraph should apply to a key finding of the study and how it fits in with the literature. I also recommend ending the paper with a very broad conclusion, not the limitations of the study.

There is interesting information in here, but I think the paper falls short of presenting it in the best way to be understood. Depending on the responses of the other reviewers, I would be willing to read further versions of this manuscript.

Respones:

We thank the reviewers for their valuable comments, and this paper has been supplemented with discussion based on their comments, and the conclusions of the study have been rewritten.

---

## [Decision Letter · Decision Letter 1]

4 Aug 2021

PONE-D-21-12141R1

Perceptions of tourists of the resources , ecological service functions and recreation value of the Guanwu National Forest Recreation Area

PLOS ONE

Dear Dr. Chen,

Thank you for submitting your manuscript to PLOS ONE. After careful consideration, we feel that it has merit but does not fully meet PLOS ONE’s publication criteria as it currently stands. Therefore, we invite you to submit a revised version of the manuscript that addresses the points raised during the review process.

We look forward to receiving your revised manuscript.

Kind regards,

Shah Md Atiqul Haq

Academic Editor

PLOS ONE

Journal Requirements:

Additional Editor Comments:

Dear authors,

I would ask you to revise the paper by following the reviewers' comments and suggestions.

Best wishes

Reviewers' comments:

Reviewer's Responses to Questions

**Comments to the Author**

1. If the authors have adequately addressed your comments raised in a previous round of review and you feel that this manuscript is now acceptable for publication, you may indicate that here to bypass the “Comments to the Author” section, enter your conflict of interest statement in the “Confidential to Editor” section, and submit your "Accept" recommendation.

Reviewer #1: (No Response)

2. Is the manuscript technically sound, and do the data support the conclusions?

Reviewer #1: Yes

3. Has the statistical analysis been performed appropriately and rigorously? 

Reviewer #1: Yes

4. Have the authors made all data underlying the findings in their manuscript fully available?

Reviewer #1: Yes

5. Is the manuscript presented in an intelligible fashion and written in standard English?

Reviewer #1: Yes

6. Review Comments to the Author

Reviewer #1: I want to thank the authors for their substantial edits to their manuscript, which I think strengthened the study overall. I admire their revision of the Results section, especially the detail devoted to making new figures depicting the results, and the addition of a Discussion section. Below, I outline several additional recommendations to continue to strengthen the manuscript:

1. Reference the supplemental information (like the survey questionnaire, the data, etc.) when it is mentioned in the manuscript. For example, you mention the survey questionnaire at lines 169-173, but do not reference where in the supplemental information the survey is. Please do this so the reader knows where to look.

2. Please explain the number of respondents you tried to interview, and the response rate of the interviews in the paragraph that begins at line 184. I know you added more references to support convenience sampling, and I appreciate that you mentioned the number of total questionnaires completed. However, an important missing piece is reporting the total number of interviews you tried to do, so we know the response rate (i.e. if you tried talking to 1000 people, but only had 500 questionnaires completed, that would be a 50% response rate). Reporting on this assists with replicability if other researchers want to do a similar study in their national park(s).

3. Please include which software you used for data processing and analysis (beginning at line 198).

4. Please include explanation of all panels in all figure descriptions. For example, Figure 1 has 10 panels and is separated into two sections (“Recreation behaviors” and “Resource evaluation”), so please describe the different panels in more detail in the figure descriptions. Figure 3 especially requires more explanation in the description as it appears to be a multidimensional plot; please describe what was displayed in the plot and what the axes represent. I have no idea how to interpret this plot otherwise.

5. I think the Discussion and Conclusions should be combined, with the details in the Conclusion included first as they summarize the key points from the study. Then, you can start to apply the main findings to the broader literature and discuss any limitations from your study. How the Discussion and Conclusions are depicted right now makes it hard to understand the key findings of the study, as they are presented after connecting it to the literature. Please rewrite. If you want to keep a Conclusions section, stick to one paragraph and summarize everything there. But only after you summarize the key findings in more detail earlier in the paper.

I see the value in getting this paper published, but there are still some limitations to me accepting it as is. I am willing to review the paper one more time if the above points are addressed. Thank you.

7. PLOS authors have the option to publish the peer review history of their article (what does this mean?). If published, this will include your full peer review and any attached files.

Reviewer #1: No

---

## [Author Response · Author response to Decision Letter 1]

21 Aug 2021

6. Review Comments to the Author

Reviewer #1: I want to thank the authors for their substantial edits to their manuscript, which I think strengthened the study overall. I admire their revision of the Results section, especially the detail devoted to making new figures depicting the results, and the addition of a Discussion section. Below, I outline several additional recommendations to continue to strengthen the manuscript:

1. Reference the supplemental information (like the survey questionnaire, the data, etc.) when it is mentioned in the manuscript. For example, you mention the survey questionnaire at lines 169-173, but do not reference where in the supplemental information the survey is. Please do this so the reader knows where to look.

Respones:

The location of the supplemental information has been added, see line 171-172.

2. Please explain the number of respondents you tried to interview, and the response rate of the interviews in the paragraph that begins at line 184. I know you added more references to support convenience sampling, and I appreciate that you mentioned the number of total questionnaires completed. However, an important missing piece is reporting the total number of interviews you tried to do, so we know the response rate (i.e. if you tried talking to 1000 people, but only had 500 questionnaires completed, that would be a 50% response rate). Reporting on this assists with replicability if other researchers want to do a similar study in their national park(s).

Respones: 

According to the statistics of interviewers, the response rate for this study was 55%, see line 192-193.

3. Please include which software you used for data processing and analysis (beginning at line 198).

Respones:

We have added the adopted software, see line 213-215.

4. Please include explanation of all panels in all figure descriptions. For example, Figure 1 has 10 panels and is separated into two sections (“Recreation behaviors” and “Resource evaluation”), so please describe the different panels in more detail in the figure descriptions. Figure 3 especially requires more explanation in the description as it appears to be a multidimensional plot; please describe what was displayed in the plot and what the axes represent. I have no idea how to interpret this plot otherwise.

Respones:

The illustrations in Figure 1 and Figure 3 have been corrected, see line 277-283 and line 350-353.

5. I think the Discussion and Conclusions should be combined, with the details in the Conclusion included first as they summarize the key points from the study. Then, you can start to apply the main findings to the broader literature and discuss any limitations from your study. How the Discussion and Conclusions are depicted right now makes it hard to understand the key findings of the study, as they are presented after connecting it to the literature. Please rewrite. If you want to keep a Conclusions section, stick to one paragraph and summarize everything there. But only after you summarize the key findings in more detail earlier in the paper.

I see the value in getting this paper published, but there are still some limitations to me accepting it as is. I am willing to review the paper one more time if the above points are addressed. Thank you.

Respones:

We still keep a Conclusions section, and strengthen the content of the Discussion section, and strengthen the linkage between Discussion and Conclusions.

---

## [Decision Letter · Decision Letter 2]

13 Sep 2021

Perceptions of tourists of the resources , ecological service functions and recreation value of the Guanwu National Forest Recreation Area

PONE-D-21-12141R2

Dear Dr. Jan-Chang Chen,

We’re pleased to inform you that your manuscript has been judged scientifically suitable for publication and will be formally accepted for publication once it meets all outstanding technical requirements.

Kind regards,

Shah Md Atiqul Haq

Academic Editor

PLOS ONE

Additional Editor Comments (optional):

Congratulations!!

The article has been accepted now.

Reviewers' comments:

Reviewer's Responses to Questions

**Comments to the Author**

1. If the authors have adequately addressed your comments raised in a previous round of review and you feel that this manuscript is now acceptable for publication, you may indicate that here to bypass the “Comments to the Author” section, enter your conflict of interest statement in the “Confidential to Editor” section, and submit your "Accept" recommendation.

Reviewer #1: All comments have been addressed

2. Is the manuscript technically sound, and do the data support the conclusions?

Reviewer #1: Yes

3. Has the statistical analysis been performed appropriately and rigorously? 

Reviewer #1: Yes

4. Have the authors made all data underlying the findings in their manuscript fully available?

Reviewer #1: Yes

5. Is the manuscript presented in an intelligible fashion and written in standard English?

Reviewer #1: Yes

6. Review Comments to the Author

Reviewer #1: Thank you for your revisions to the paper. I am looking forward to seeing it published soon. Please be sure to include high-definition versions of all figures, though, as some of them are a bit blurry in your submission.

7. PLOS authors have the option to publish the peer review history of their article (what does this mean?). If published, this will include your full peer review and any attached files.

Reviewer #1: No

---

## [Editor Report · Acceptance letter]

22 Sep 2021

PONE-D-21-12141R2 

Perceptions of tourists of the resources, ecological service functions and recreation value of the Guanwu National Forest Recreation Area 

Dear Dr. Chen:

I'm pleased to inform you that your manuscript has been deemed suitable for publication in PLOS ONE. Congratulations! Your manuscript is now with our production department. 

Kind regards, 

on behalf of

Dr. Shah Md Atiqul Haq 

Academic Editor

PLOS ONE